# Integrating Satellite and UAV Technologies for Maize Plant Height Estimation Using Advanced Machine Learning

**Marcelo Araújo Junqueira Ferraz** *[ID], **Thiago Orlando Costa Barboza** [ID], **Pablo de Sousa Arantes**, **Renzo Garcia Von Pinho** and **Adão Felipe dos Santos** *[ID]

Department of Agriculture, School of Agricultural Sciences of Lavras, Federal University of Lavras (UFLA), Lavras 37200-900, Brazil; thiago.barboza1@estudante.ufla.br (T.O.C.B.); pablo.arantes2@estudante.ufla.br (P.S.A.); renzo@ufla.br (R.G.V.P.)

* Correspondence: marcelo.ferraz1@estudante.ufla.br (M.A.J.F.); adao.felipe@ufla.br (A.F.S.)

**Abstract:** The integration of aerial monitoring, utilizing both unmanned aerial vehicles (UAVs) and satellites, alongside sophisticated machine learning algorithms, has witnessed a burgeoning prevalence within contemporary agricultural frameworks. This study endeavors to systematically explore the inherent potential encapsulated in high-resolution satellite imagery, concomitantly accompanied by an RGB camera seamlessly integrated into an UAV. The overarching objective is to elucidate the viability of this technological amalgamation for accurate maize plant height estimation, facilitated by the application of advanced machine learning algorithms. The research involves the computation of key vegetation indices—NDVI, NDRE, and GNDVI—extracted from PlanetScope satellite images. Concurrently, UAV-based plant height estimation is executed using digital elevation models (DEMs). Data acquisition encompasses images captured on days 20, 29, 37, 44, 50, 61, and 71 post-sowing. The study yields compelling results: (1) Maize plant height, derived from DEMs, demonstrates a robust correlation with manual field measurements ($r = 0.96$) and establishes noteworthy associations with NDVI ($r = 0.80$), NDRE ($r = 0.78$), and GNDVI ($r = 0.81$). (2) The random forest (RF) model emerges as the frontrunner, displaying the most pronounced correlations between observed and estimated height values ($r = 0.99$). Additionally, the RF model's superiority extends to performance metrics when fueled by input parameters, NDVI, NDRE, and GNDVI. This research underscores the transformative potential of combining satellite imagery, UAV technology, and machine learning for precision agriculture and maize plant height estimation.

**Keywords:** precision agriculture; artificial intelligence; plant growth; random forest; KNN

## 1. Introduction

Maize (*Zea mays* L.) stands as one of the foremost globally cultivated and adaptable crops, serving as a nutritional sustenance for both human populations and animals [1]. Nevertheless, variables like canopy vitality, nutritional status, adaptability, and plant growth exert substantial influence on ultimate yield outcomes. The interplay of these factors, coupled with environmental and genetic diversity, can wield significant sway over plant height dynamics [2].

Efficient and accurate assessment of plant height is paramount in appraising maize's growth potential, furnishing agronomists with essential insights into plant development for well-informed decision-making regarding field management practices. In recent years, innovative methodologies encompassing remote sensing techniques, unmanned aerial vehicle (UAV) imagery, and the power of machine learning (ML) have been steadily gaining prominence in modern agricultural paradigms [3–6]. In this context, achieving swift and accurate large-scale estimations of maize plant height and enabling dynamic growth monitoring [7] play a pivotal role in amplifying crop management strategies [8], facilitating evaluations of cultivars in the fields, and empowering informed decision-making among agricultural stakeholders.

In recent studies, UAVs equipped with red-green-blue (RGB) sensors [9,10], multi-spectral sensors [11], hyperspectral sensors [12], and LiDAR systems [13] have been utilized to estimate plant height in various crops. The adoption of RGB cameras is particularly interesting due to their accessibility and operational simplicity [8], enabling the generation of digital surface models that provide insights into vegetation height [14,15]. However, the underexplored potential in vegetation indices, which directly correlate with canopy structural inputs and spectral responses [16], has been relatively underutilized in maize plant height estimation. Another significant advancement in agriculture is the integration of machine learning, which has substantially enhanced the processing of extensive data sets and demonstrated precision in estimating critical agronomic parameters [6,17,18].

Numerous regression techniques, including conventional linear regression, can encounter challenges when applied to a specific set of input data due to data complexity and multicollinearity among predictor variables [19,20]. In contrast, regressions founded on machine learning algorithms, such as random forest (RF), K-nearest neighbor (KNN), support vector machine (SVM), and decision trees (DT), offer enhanced precision and are tailored to address intricate interactions. These algorithms prove valuable in estimating critical agronomic parameters across diverse crops [2,21–24]. Nonetheless, machine learning algorithms rely on a single estimation model and may tend to overfit when confronted with limited training data, as observed with KNN, SVM, and RF [8]. To alleviate such concerns, pre-processing input data through normalization, utilizing extensive databases [25], and applying data partitioning methods like K-fold [26] can effectively mitigate the risks of overfitting and underfitting during the training process, thereby bolstering the models' capacity for generalization. Integrating machine learning with data from UAVs and satellites stands as a promising strategy for monitoring plant height dynamics at a large experimental scale. This method not only meets the demand for remote assessment but also provides cost-effective implementation, enhanced flexibility, reduced labor, and heightened precision.

This study leveraged vegetation indices extracted from PlanetScope images and digital elevation models generated from UAV imagery. These data were harnessed alongside machine learning algorithms implemented in Python to accurately estimate plant height within field settings. The study aimed to achieve three key objectives: (1) to employ machine learning for plant height estimation; (2) to evaluate the viability of vegetation indices and digital elevation models as input parameters for machine learning algorithms; and (3) to ascertain the most effective algorithm for precise plant height estimation.

## 2. Materials and Methods

Figure 1 provides a graphical overview of the workflow utilized in this study, incorporating an UAV equipped with an RGB sensor and high-resolution satellite imagery. The acquisition of maize crop images coincided with on-site field assessments, and UAV flights were strategically conducted around midday to reduce potential image distortions. Leveraging the UAV data, digital elevation models were generated to derive plant heights, while vegetation indices were extracted from orbital imagery from PlanetScope. Subsequently, a machine learning model was implemented to facilitate the estimation of plant height within the field context.

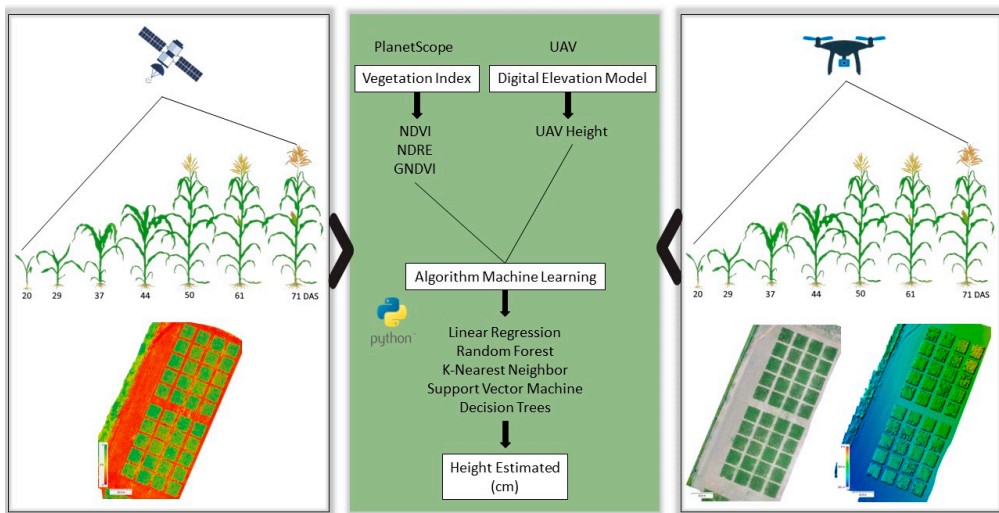

**Figure 1.** Summary of the study to estimate plant height using VI and DEM based on UAV in maize.

## 2.1. Study Area

The study was conducted in Ijaci, Minas Gerais, Brazil, situated at coordinates 21°09′40″ S, 44°55′03″ W (Figure 2). The region inputs a subtropical climate classified as Cwa, characterized by dry winters and warm summers, with an average temperature of 20.9 °C and an annual precipitation of 1325 mm according to classification [27]. The study site is located at an average elevation of 842 m. The research focused on data collected during the agricultural season of 2021/2022, specifically targeting the maize crop (*Zea mays* L.). The experimental area was subdivided into 40 plots, each measuring 10 × 10 m (100 m$^2$). The maize was sown on 11 October 2021 using an early cycle variety with semi-erect leaves.

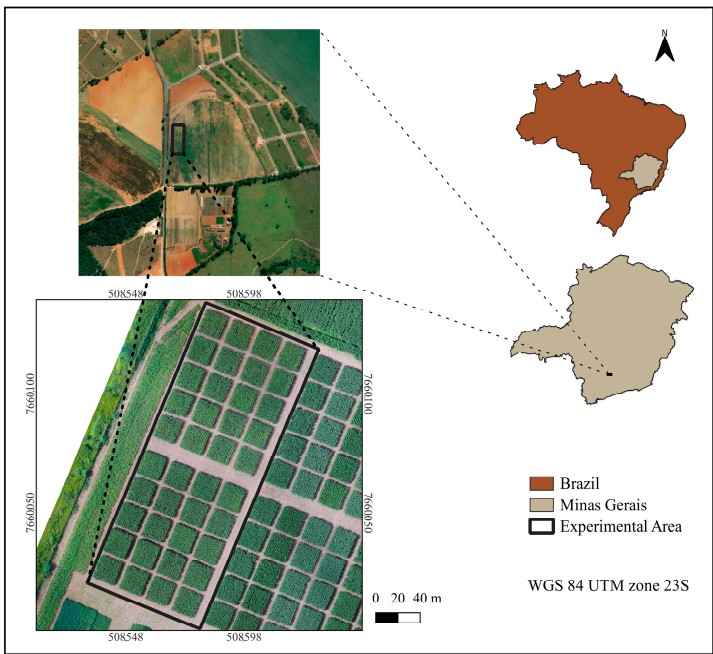

**Figure 2.** Location of the experimental area in Ijaci, MG.

## 2.2. Data Acquisition in the Field

Field measurements of plant height were conducted manually using a measuring tape at specific intervals: 20, 29, 37, 44, 50, 61, and 71 days after sowing (DAS). The imaging captures were conducted on 1, 9, 17, 24, and 30 November, as well as on 11 and 21 December 2022. The experimental plots comprised 16 rows with a row-to-row

spacing of 60 cm. To minimize border effects, height measurements were focused on the central four rows. The measurements were taken for 15 individual plants within each experimental plot, from the soil surface up to the insertion point of the flag leaf [28]. This methodology yielded a substantial dataset, comprising 600 samples for each observation date across the 40 experimental plots. In total, 4200 samples were collected, spanning from the emergence of the crop to its initial reproductive stage (R1). These comprehensive assessments were conducted not only to monitor the growth progression of maize but also to validate the viability of employing a vegetation index and digital elevation model for the precise estimation of plant height by implementing machine learning methodologies.

### 2.3. Image Acquisition and Processing

The images were acquired using two platforms: a Phantom 4 UAV (SZ DJI Technology Co., Shenzhen, China) equipped with an RGB camera (model FC330, DJI, Shenzhen, China), and the sensor from the PlanetScope CubeSat platform (Planet Labs Inc., San Francisco, CA, USA) for satellite images. Orbital imagery was harnessed to extract vegetation indices, whereas UAV imagery was used to generate digital elevation models.

Both the UAV flights and the acquisition of satellite images were synchronized with manual field assessments of plant height. The UAV flight plan was meticulously orchestrated using Pix4D Capture software (Version 4.13.1, Pix4d SA, Prilly, Switzerland), incorporating an 80% frontal and 75% lateral overlap, with a flight altitude of 40 m. The RGB camera, oriented orthogonally to the ground, yielded a ground sample distance (GSD) of 1.09 cm per pixel. Nine ground control points (GCPs) were strategically positioned within the study area to ensure precise geographic referencing [29], thereby augmenting the fidelity of crop information extraction. Geographical coordinates for these points were gathered using GPS equipment, with Real Time Kinematic (RTK) signal-correction-enhancing accuracy.

Digital elevation model generation entailed Pix4D Mapper software (Version 4.5, Pix4d SA, Prilly, Switzerland), encompassing a systematic workflow encompassing image alignment, dense point cloud generation, orthomosaic development, and the creation of digital surface models for each assessment date.

In relation to the orbital imagery, the PlanetScope platform facilitated daily data acquisition, characterized by a spatial resolution range of 3 to 5 m, encompassing eight spectral bands: Coastal Blue (431–452 nm), Blue (465–515 nm), Green I (513–549 nm), Green (547–583 nm), Yellow (600–620 nm), Red (650–680 nm), Rededge (697–713 nm), and Near-infrared (NIR) (845–885 nm) [30]. The platform offered the PlanetScope analytic ortho scene surface reflectance (SR—Level 3B) product, ensuring orthorectification, geometric and radiometric correction, Universal Traverse Mercator (UTM) projection, and atmospheric radiance calibration. The resultant imagery was provided with radiance, surface reflectance, and a GeoTiff format [31]. To reduce the impact of varying weather conditions, particularly cloud cover, PlanetScope images were selected within a maximum interval of three days variation.

### 2.4. Extraction of Vegetation Indices and Digital Elevation Model

The vegetation indices were derived from reflectance values obtained from PlanetScope orbital images (Table 1) using the QGIS software (Version 3.22.15, QGIS Development Team, Trondheim, Norway). To minimize edge effects on plant reflectance, a negative buffer of 1.0 m was applied within each 100 $m^2$ parcel. Subsequently, 15 random sampling points were generated within this buffer area, and the Point Sampling Tool (Version 0.5.4) [32] plugin was utilized to associate raster values of each vegetation index with the sampled points in the parcels. The selection of these indices was motivated by their established correlation with key biophysical traits of crops, including biomass, canopy health, and chlorophyll content [33,34].

**Table 1.** Vegetation indices for PlanetScope orbital images.

| VI | Equation | Reference |
|---|---|---|
| NDVI [1] | (NIR—Red)/(NIR + Red) | [35] |
| NDRE | (NIR—Rededge)/(NIR + Rededge) | [36] |
| GNDVI | (NIR—Green)/(NIR + Green) | [37] |

[1] NDVI: normalized difference vegetation index; NDRE: normalized difference red edge index; GNDVI: green normalized difference vegetation index.

To derive plant height values from UAV data, the digital elevation model (DEM) (Equation (1)) was computed for each assessment date. The initial UAV flight performed on the day of maize sowing was utilized as the reference digital terrain model (DTM). For the subsequent time points (20, 29, 37, 44, 50, 61, and 71 days after sowing), image processing led to the creation of digital surface models (DSMs) [14,15,38–40]. These DSMs were then input into QGIS software, where Equation (1) was applied to generate the DEM values corresponding to each assessment date.

$$DEM = DSM - DTM \tag{1}$$

The segmentation of the ground and vegetation portions from the orthomosaic was conducted through a supervised classification approach using the Dzetsaka Classification Tool plugin [41] within the QGIS software, following the methodology utilized in the study by [42]. Subsequently, focusing solely on the vegetative component, a vectorized representation was generated, which in turn was employed to extract the corresponding section from the raster DEM, effectively eliminating the ground component from the digital elevation model. Within the confines of each experimental plot, the identical set of 15 points was consistently employed to facilitate the acquisition of height values by utilizing the Point Sampling Tool plugin.

### 2.5. Machine Learning Algorithms for Plant Height Estimation

The study employed a range of machine learning algorithms to estimate field plant height, utilizing inputs such as vegetation indices (NDVI, NDRE, and GNDVI) and UAV-derived plant height (UAV height) from the DEM. The algorithms evaluated included linear regression (LR) [43], random forest (RF) [44], K-nearest neighbor (KNN) [45], support vector machine (SVM) [46], and decision trees (DT) [47]. The application of these machine learning algorithms in the agricultural sector is commonplace owing to their adaptability to complex datasets, capacity to capture non-linear interrelationships among variables, and aptitude for generalizing acquired patterns to novel datasets.

To ensure robust analysis, three distinct data input configurations were established: Input 1 (UAV height), Input 2 (UAV height + NDVI + NDRE + GNDVI), and Input 3 (NDVI + NDRE + GNDVI). Cross-validation using the K-fold method was employed, distributing the dataset into K subgroups for thorough comparison, a strategy to enhance estimation precision while minimizing the risk of overfitting [48].

The K-fold technique divided the data randomly into K subsets of equal size, with each subset being used for training and validation. In our study, K was set to five, leading to five iterations of training and validation. Implementation of these machine learning algorithms was carried out using Python (Version 3.11, PYTHON Software Foundation).

### 2.6. Pre-Processing of Data and Statistical Analysis

Data normalization is a pivotal pre-processing step that aims to harmonize variables with varying magnitudes and distributions. Its purpose is to ensure uniformity in scale and value distribution across all variables, thereby promoting equitable contributions to model development and diminishing potential bias stemming from variable dominance. In this context, the data underwent a transformation to attain a mean of zero and a stan-

dard deviation of 1, employing the StandardScaler method available in the Scikit-learn library [25,49,50].

To assess the efficacy of algorithms in estimating plant height, two statistical metrics were employed: the coefficient of determination ($R^2$) (Equation (2)) and root-mean-square error (RMSE) (Equation (3)). All statistical analyses conducted throughout this study were executed using Python 3.11.

$$R^2 = 1 - \frac{\sum_{i=1}^{n}(\mathrm{yi} - \overline{y}\mathrm{i})^2}{\sum_{i=1}^{n}(\mathrm{yi} - \hat{y})^2} \tag{2}$$

$$\mathrm{RMSE} = \sqrt{\frac{1}{n}\sum_{i=1}^{n}(\mathrm{yi} - \overline{y}\mathrm{i})^2} \tag{3}$$

## 3. Results

### 3.1. Correlation between Observed and Estimated Height Values

The Pearson correlation coefficient ($r$) was employed to assess the relationships between data acquired from both the UAV and satellite platforms, revealing a positive correlation with manually collected plant height measurements in the field. The estimated plant heights generated by the models exhibited statistically significant correlations ($p < 0.05$) across all three input configurations, with particularly relevant outcomes observed in the RF and KNN models (Figure 3).

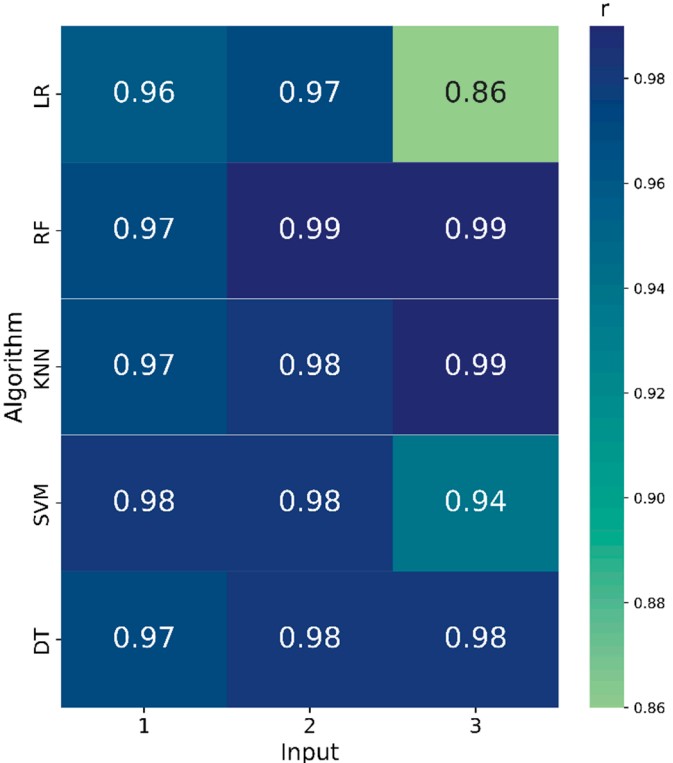

**Figure 3.** Pearson correlation coefficients ($r$) between observed and estimated values of plant height for each algorithm and input.

The Pearson coefficient ranged from 0.96 to 0.98 in the first input configuration and from 0.97 to 0.99 in the second input while varying from 0.86 to 0.99 for the third input configuration. The correlation values demonstrated consistency among the RF, KNN, and DT models, with SVM models displaying a slight reduction in correlation ($r = 0.94$) and LR models yielding the lowest correlation values ($r = 0.86$). The linear regression model is well-suited for scenarios featuring linear relationships in the data, and its reduced precision in input configuration 3 could be attributed to the presence of vegetation indices.

Significantly, the RF (*r* = 0.99) and KNN (*r* = 0.99) models exhibited the strongest correlations in input configuration 3. Overall, the correlation analysis underscores the superior performance of RF models in estimating field plant height. By scrutinizing the linear dependence between estimated and observed height values, it becomes evident that the most effective models for accurate plant height estimation leverage UAV-derived height data and vegetation indices as input parameters for the machine learning process.

### 3.2. Comparison and Performance of Machine Learning Algorithms

To evaluate the effectiveness of machine learning algorithms in estimating field plant height, three distinct input configurations (Inputs 1, 2, and 3) were established and evaluated (Table 2). Incorporating both UAV height and vegetation indices as input variables notably enhanced model performance. Among the machine learning approaches, the RF and KNN models emerged as superior performers, exhibiting heightened precision and accuracy compared to other algorithms for plant height estimation, albeit with minor variance in RMSE.

**Table 2.** Precision ($R^2$) and accuracy (RMSE) of training and testing models for estimating plant height in maize.

| Algorithms | Input [1] | Training | | Test | |
|---|---|---|---|---|---|
| | | $R^2$ | RMSE (cm) | $R^2$ | RMSE (cm) |
| Linear Regression | 1 | 0.93 | 24.56 | 0.93 | 23.56 |
| | 2 | 0.93 | 22.71 | 0.94 | 21.33 |
| | 3 | 0.74 | 45.01 | 0.74 | 44.13 |
| Random Forest | 1 | 0.94 | 24.49 | 0.94 | 22.02 |
| | 2 | 0.97 | 16.49 | 0.97 | 15.07 |
| | 3 | 0.97 | 15.76 | 0.97 | 14.62 |
| K-Nearest Neighbor | 1 | 0.95 | 18.74 | 0.95 | 20.59 |
| | 2 | 0.97 | 14.10 | 0.97 | 16.55 |
| | 3 | 0.97 | 11.84 | 0.97 | 14.66 |
| Support Vector Machine | 1 | 0.95 | 17.81 | 0.95 | 19.39 |
| | 2 | 0.95 | 15.86 | 0.95 | 18.76 |
| | 3 | 0.87 | 30.86 | 0.88 | 32.02 |
| Decision Trees | 1 | 0.94 | 19.76 | 0.94 | 22.29 |
| | 2 | 0.98 | 15.60 | 0.97 | 16.98 |
| | 3 | 0.97 | 14.84 | 0.97 | 16.26 |

[1] Input 1: UAV height; Input 2: UAV height + NDVI + NDRE + GNDVI; Input 3: NDVI + NDRE + GNDVI.

Specifically, the SVM model ($R^2$ = 0.95 and RMSE = 19.39 cm) demonstrated superior accuracy with Input 1, while the RF model displayed the highest accuracy for Inputs 2 and 3. The DT model exhibited performance akin to RF and KNN and notably improved accuracy with Input 3 ($R^2$ = 0.97 and RMSE = 16.26 cm). Conversely, the conventional linear regression model exhibited suboptimal accuracy and precision with Input 3 due to the inherent limitation of a linear trend line to effectively capture the complexities inherent in vegetation index data. Notably, in this investigation, models derived from RF, KNN, and DT algorithms showcased greater accuracy and precision in estimating field plant height, integrating insights from orbital remote sensing and UAV-derived digital models. Remarkably, the RF and KNN models displayed minimal errors when utilizing Input 3.

Furthermore, assessing the models' performance in estimating maize plant height using three input configurations involved analyzing RMSE and $R^2$ values (Figure 4). The RF and KNN models exhibited superior performance with input configuration 3, highlighting the efficacy of incorporating vegetation indices (NDVI, NDRE, and GNDVI) in accurately estimating maize plant height. Input configuration 1 (UAV Height) demonstrated optimal performance with the SVM model, whereas Input 2 (UAV height + NDVI + NDRE + GNDVI) yielded favorable outcomes with the RF model.

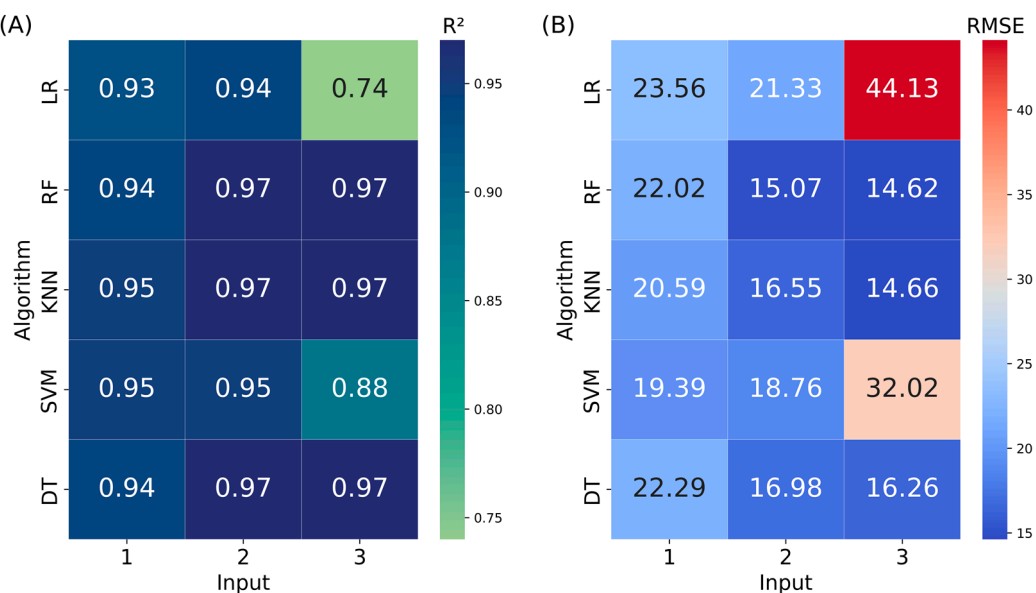

**Figure 4.** Coefficient of determination ($R^2$) (**A**) and root-mean-square error (RMSE) (**B**) in centimeters of the ML algorithms implemented in the study.

The RF model consistently demonstrated superior precision and accuracy across input configurations 3 and 2 ($R^2$ = 0.97), resulting in RMSE values of 14.62 and 15.07 cm, respectively. Similarly, the KNN model exhibited comparable performance with Input 3, yielding an $R^2$ value of 0.97 and an RMSE of 14.66 cm. These outcomes underscore the feasibility of leveraging vegetation indices in conjunction with ML algorithms for accurate maize plant height estimation. Contrastingly, the linear regression models yielded good results for inputs 1 ($R^2$ = 0.93) and 2 ($R^2$ = 0.94), particularly considering their reliance on UAV-based height as an input feature. However, in the context of input configuration 3, encompassing all three vegetation indexes, the linear regression model exhibited reduced precision, obtaining an $R^2$ value of 0.74 and an RMSE of 44.13 cm. This decline in accuracy, amounting to a 21% reduction, underscores the limited suitability of the traditional linear regression approach for capturing non-linear data trends, a phenomenon confirmed by earlier findings such as those of [51]. Furthermore, the superior precision demonstrated by the RF, SVM, and KNN models over linear regression can be attributed to their adeptness at effectively extracting insights from datasets characterized by non-linearity and multicollinearity among predictor variables.

### 3.3. Estimation of Plant Height in the Field

Considering the precision and accuracy values of the plant height estimation models based on the four input variables (UAV height, NDVI, NDRE, and GNDVI), it is evident that the RF and KNN algorithms exhibit prominent similarities and a notable ability to estimate plant height in the field, while maintaining comparable errors (Figure 5). The RF model was the best in terms of performance for input configurations 2 and 3. Similarly, the KNN model demonstrated a precision close to that of RF for input configuration 3. A noteworthy aspect is the improvement in precision and accuracy compared to estimator models, as exemplified by the RF model achievement of $R^2$ = 0.97 and RMSE = 14.62 cm. Moreover, these algorithms showcased analogous performance when leveraging the vegetation indices (Input 3), displaying only a 0.04 cm RMSE difference between the two.

Broadening our perspective, the observed plant height 71 days after sowing (R1 stage) in the field stood at 255.48 cm. Conversely, the height estimated using the RF and KNN models was recorded at 255.39 cm and 256.89 cm, respectively. This comprehensive analysis (Figure 5) underscores that estimation models using the NDVI, NDRE, and GNDVI variables achieved a higher precision and accuracy compared to their counterparts. As such, the estimation of maize plant height proves to be effectively conducted through the

adept utilization of the RF and KNN algorithms through a general model, particularly during the R1 stage, characterized by the plant's attainment of maximal growth potential.

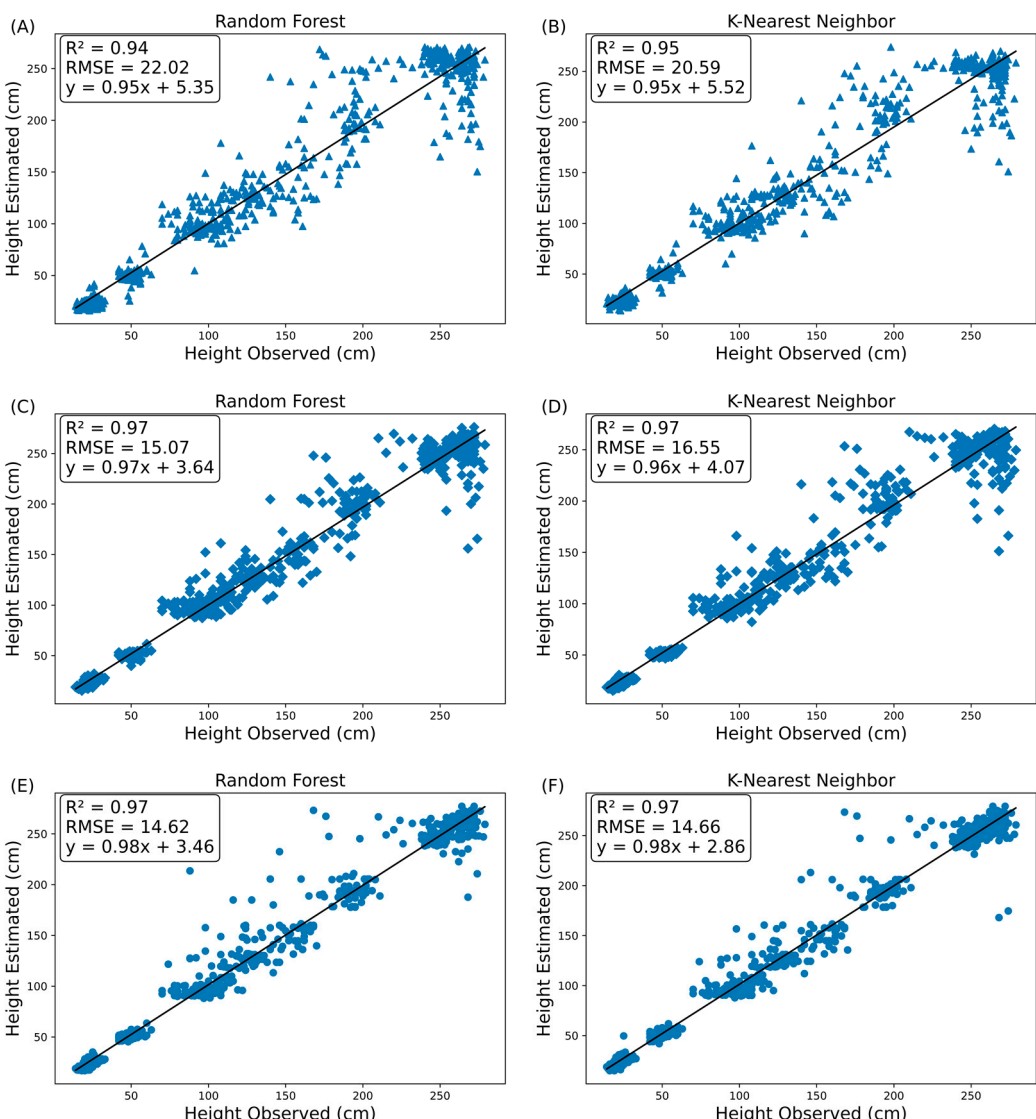

**Figure 5.** Plant height estimation for RF and KNN models. These refer to the input configurations: (1) UAV height (**A**,**B**), (2) UAV height + NDVI + NDRE + GNDVI (**C**,**D**), and (3) NDVI + NDRE + GNDVI (**E**,**F**).

## 4. Discussion

In recent years, the utilization of UAV imagery, orbital remote sensing, vegetation indices, and machine learning techniques has garnered significant attention as a technology capable of reducing the need for destructive assessments. This approach not only saves time and effort but also proves interesting for monitoring the growth dynamics of crops [7,17,40,52].

Machine learning algorithms, including RF, KNN, SVM, and DT, have emerged as promising tools for optimizing plant height estimation. This is especially relevant in large-scale maize fields, where the efficient use of resources is paramount. The dataset employed encompassed height information derived from UAV-based digital elevation models and vegetation indices obtained from the PlanetScope satellite, collected at intervals of 20, 29, 37, 44, 50, 61, and 71 days following crop sowing.

The resulting models, generated from three distinct input configurations (Input 1, 2, and 3), underwent evaluation based on R-squared values, root-mean-square error, and

Pearson correlation coefficients between observed and estimated plant height. The RF model performed better ($R^2$ = 0.97, RMSE = 14.62 cm), closely followed by the KNN model ($R^2$ = 0.97, RMSE = 14.66 cm) (Figure 4), leveraging the NDVI, NDRE, and GNDVI vegetation indices as input parameters. This underscores the pivotal role of vegetation indices in enhancing the accuracy of estimating this critical agronomic variable [11].

The RF model demonstrated the highest degree of accuracy (RMSE = 14.66 cm) compared to other algorithms, highlighting the necessity of fomenting a diversified dataset to enhance the model's robustness and generalizability. This study revealed that both the RF and KNN algorithms yielded precise outcomes for height estimation, which aligns with findings from other scholarly inquiries that harnessed these algorithms to estimate the heights of maize and soybean crops using LiDAR and UAV data [2], maize height estimation via a three-dimensional model [9], estimation of summer maize growth using digital models [53], biomass estimation based on UAV data [54], determination of leaf area index [1], and estimation of bean production using UAV RGB images [8]. Furthermore, in addition to the RF algorithm, the integration of the multilayer perceptron (MLP) shows promise in estimating plant height based on UAV RGB images [7].

UAVs have garnered widespread attention as remote sensing platforms owing to their inherent advantages, including flexibility, high spatial and temporal resolution, extensive overlap rates, cost-effectiveness, and enhanced accessibility [55]. As a result, this study proposes the extraction of height information using UAV-based digital elevation models on seven different periods, thereby encompassing data from the early vegetative stages to the R1 stage.

Photogrammetric techniques are employed to construct digital models of the Earth's surface and perform precise geometric measurements from UAV images, as noted by [10,12,56,57]. After the aerial image processing, DEM is derived by calculating the difference between the DSM and DTM. This DEM is then utilized to extract essential information about the height of maize crops. The results of this study's plant height estimations align with the findings of [15,58], illustrating that the DEM serves as a consistent and reliable variable for vegetation height estimation. However, a disparity emerges during the early growth stages of maize, where field-measured plant height surpasses the height extracted from the DEM.

This discrepancy could potentially be attributed to the methodology employed in manual assessments, which considers canopy height as the vertical distance between the insertion point of the flag leaf and the ground surface. Moreover, the dense point cloud generated from UAV images may capture a comprehensive range of height information on the surface, including lower morphological structures of the plant. This observation was also found by [59], who employed UAV RGB images to estimate maize plant heights, indicating that the estimated values were relatively lower when contrasted with field observations. This discrepancy could stem from the limited accuracy of the dense point cloud in capturing the uppermost point of the maize plant, as indicated by [60]. To enhance the quality of information extracted from the DEM, a potential solution is to strategically distribute more ground control points within the study area, thereby mitigating model errors and refining spatial accuracy, as suggested by [29].

It is crucial to underscore that in this study, during the initial developmental phases of the crop, UAV-derived plant height values did not precisely mirror the field-observed heights, signifying that the general model may exhibit limited efficacy when applied in the early stages of maize growth. However, as the crop progresses to later stages, up to the reproductive stage R1, the model precision and accuracy in height estimation tend to improve. This is underpinned by the comprehensive data collection period, spanning from 20 to 71 days after sowing. This trend aligns with the findings of [39], where the model exhibited superior performance when utilizing data collected up to and encompassing the flowering stage, estimating black oat height through digital surface models and RGB-based vegetation indices. Additionally, the absence of full canopy closure during the early stages might account for underestimated plant height values, as the digital model must factor

in information at the leaf level. Thus, images with lower spatial resolutions might yield values that do not adequately represent the area, as elucidated by [61].

The considerable challenge in estimating plant height on a large scale necessitates the development of an efficient and accurate methodology. In response, this study proposed an innovative approach by integrating vegetation index data, digital elevation models, and a range of machine learning algorithms for precise plant height estimation. The outcomes derived from the RF model ($R^2$ = 0.97, RMSE = 14.62 cm, and $r$ = 0.99) offer a promising avenue to potentially supplant traditional manual field measurements, commonly executed using rudimentary tools like rulers or tape measures, particularly for short-statured commercial crops such as maize, soybean, rice, and wheat [58].

When concentrating solely on vegetation index data (Input 3) as inputs for the models, both the RF and KNN models exhibited lower RMSE values than the other two input configurations. This pattern aligns with and extends the results found in [11], where the RF (RMSE = 16.7 cm) and KNN (RMSE = 19.4 cm) models displayed strong performance when utilizing only the vegetation index dataset. Similarly, this trend is consistent with the research conducted by [62], who investigated soybean plant height estimation using machine learning techniques. Their work reinforces the superiority of the RF model over other algorithms (SVM and LR) when incorporating solely vegetation indices as input parameters, a framework consistent with the NDVI, NDRE, and GNDVI indices employed in our present study.

Hence, upon combining UAV-derived plant height with vegetation indices (Input 2), a light increase of 3% and 11% in RMSE values was noted for the RF and KNN algorithms, respectively (Figure 5). Correspondingly, a decline of 33% (RF) and 29% (KNN) in accuracy was observed when solely utilizing UAV plant height as the input (Input 1). The RF and KNN algorithms exhibited considerable performance owing to their adeptness in handling complex and non-linear data patterns. Particularly, RF demonstrated reduced susceptibility to overfitting [23].

The outcomes derived from the linear regression models outperformed those reported by [63], who employed UAV imagery and linear regression for field measurements ($R^2$ = 0.88) and established a robust correlation between observed and estimated heights. Nevertheless, when incorporating vegetation indices as inputs in the present study, LR exhibited the lowest precision and accuracy ($R^2$ = 0.74 and RMSE = 44.13 cm) relative to the other algorithms. This divergence could be attributed to the inherent rigidity of the LR model with respect to non-parametric data, thereby impeding its capacity to capture the nuanced non-linearity and intricacy of the data [14], ultimately leading to suboptimal performance in height estimation.

## 5. Conclusions

In a broader context, this study has effectively demonstrated the applicability of UAV imagery, orbital remote sensing, and machine learning algorithms to accurately estimate plant height within field conditions. The investigation focused on two specific machine learning algorithms, random forest, and k-nearest neighbors, showcasing better performance. Notably, the RF algorithm exhibited a superior outcome ($R^2$ = 0.97, RMSE = 14.62 cm), closely followed by the KNN algorithm ($R^2$ = 0.97, RMSE = 14.66 cm) when utilizing the vegetation indices NDVI, NDRE, and GNDVI as inputs parameters.

This trend persisted when incorporating UAV height in conjunction with the aforementioned vegetation indices, where the RF algorithm showcased commendable precision and accuracy ($R^2$ = 0.97, RMSE = 15.07 cm). An important observation emerges from the significant correlation of the derived NDVI, NDRE, and GNDVI indices from PlanetScope imagery with maize plant height ($r$ = 0.80, $r$ = 0.78, and $r$ = 0.81, respectively), further reinforcing the utility of these indices in height estimation.

**Author Contributions:** Conceptualization, M.A.J.F. and A.F.S.; methodology, M.A.J.F.; software, M.A.J.F.; validation, M.A.J.F. and A.F.S.; formal analysis, M.A.J.F., R.G.V.P. and A.F.S.; investigation, M.A.J.F. and A.F.S.; resources, R.G.V.P. and A.F.S.; data curation, M.A.J.F. and A.F.S.; writing—original draft preparation, M.A.J.F.; writing—review and editing, M.A.J.F. and A.F.S.; visualization, T.O.C.B., P.S.A., R.G.V.P. and A.F.S.; supervision, A.F.S.; project administration, R.G.V.P.; funding acquisition, R.G.V.P. All authors have read and agreed to the published version of the manuscript.

**Funding:** This research was funded by the Research Support Foundation of the State of Minas Gerais (FAPEMIG) through project 11680.

**Data Availability Statement:** Data are contained within the article.

**Acknowledgments:** The authors are grateful to the Federal University of Lavras for all its support of this research, and for the Research Support Foundation of the State of Minas Gerais (FAPEMIG), and the National Council for Scientific and Technological Development (CNPq) for the students' scholarship.

**Conflicts of Interest:** The authors declare no conflicts of interest.

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
