# Peer review of "Integrating Satellite and UAV Technologies for Maize Plant Height Estimation Using Advanced Machine Learning"

_agriengineering, doi:10.3390/agriengineering6010002_

Round 1
Reviewer 1 Report
Comments and Suggestions for Authors
The article “Integrating Satellite and UAV Technologies for Maize Plant Height Estimation through Advanced Machine Learning” is aimed at a systematic study of the potential contained in high-resolution satellite images and their application in modern agricultural systems. The main purpose of the research is to find out the viability of a technological amalgamation for accurate maize plant height estimation, which is facilitated by the use of advanced machine learning algorithms.
The research includes the study of key vegetation indices – NDVI, NDRE and GNDVI - improved due to satellite images of the PlanetScope. At the same time, the height of plants is estimated on the basis of unmanned aerial vehicles using digital elevation models (DEMs). The data collection includes images taken over several days 20, 29, 37, 44, 50, 61, and 71 after sowing.
The authors gave a description of a real research. The study was conducted in Ijaci, Minas Gerais, Brazil, situated at coordinates 21°09’40” S, 44°55’03” W. The study site is located at an average elevation of 842 m. The research focused on data collected during the agricultural season of 2021/2022.
The study focused on specific machine learning algorithms: Random Forest and k-Nearest Neighbors, demonstrating comparable performance under different input data configurations in terms of compliance metrics such as R-squared, RMS error (RMSE) and correlation coefficient (r) between observed and calculated plant height values.
A comparative analysis of the field measurements of plant height with the results obtained using machine learning methods was carried out. HotMap-diagrams are constructed showing the effectiveness of models in estimating the height of maize plants.
Because of the study, it was possible to effectively demonstrate the applicability of images from unmanned aerial vehicles (UAV), orbital remote sensing and various machine learning algorithms for accurate assessment of plant height in the field. Moreover, the obtained results emphasize the feasibility of using an UAV with an RGB camera as a promising alternative for manually assessing the height of plants in the field.
The relevance of the research is beyond doubt, but the authors have not disclosed it enough in their work. Recently, the use of aerial monitoring using both unmanned aerial vehicles and satellites, along with complex machine learning algorithms, has become increasingly used in modern agricultural systems.
The article is well structured, with a clearly described methodology, the main part, and the results. The authors present the results of their experiments in a clear and understandable form, with tables and figures that effectively illustrate the main conclusions. The presented graphs and diagrams allow to visually evaluating the results of the experiment. The references, which includes 64 sources, effectively prepares the ground for the authors' own research and describes in detail the problematic situations related to this topic. High originality of the text of the article, which shows the high personal contribution of the authors.
General comments on the article.
1) It is necessary to better reveal the relevance of the research and show the scientific novelty of the work.
2) To conduct a more reasoned justification of the selected machine learning methods, the reasons for choosing these methods.
3) Make a comparison with similar research; provide references to them.
4) There are many abbreviations in the text that will be incomprehensible to readers who are not familiar with the topic of the research. It is necessary to provide a transcript at the first mention of abbreviations.
Despite the comments, the study is a valuable contribution to the field of studying aerial monitoring using UAVs and satellites in agricultural systems, as the authors' work is well done and the results are clearly presented. The article notes the transformative potential of combining satellite imagery, unmanned aerial vehicle technologies and machine learning for precision farming and estimating the height of maize plants. The presented technological approach provides valuable opportunities for both producers and researchers to improve monitoring of the dynamics of maize crop growth.
Author Response
The article “Integrating Satellite and UAV Technologies for Maize Plant Height Estimation through Advanced Machine Learning” is aimed at a systematic study of the potential contained in high-resolution satellite images and their application in modern agricultural systems. The main purpose of the research is to find out the viability of a technological amalgamation for accurate maize plant height estimation, which is facilitated by the use of advanced machine learning algorithms. The research includes the study of key vegetation indices – NDVI, NDRE and GNDVI - improved due to satellite images of the PlanetScope. At the same time, the height of plants is estimated on the basis of unmanned aerial vehicles using digital elevation models (DEMs). The data collection includes images taken over several days 20, 29, 37, 44, 50, 61, and 71 after sowing. The authors gave a description of a real research. The study was conducted in Ijaci, Minas Gerais, Brazil, situated at coordinates 21°09’40” S, 44°55’03” W. The study site is located at an average elevation of 842 m. The research focused on data collected during the agricultural season of 2021/2022. The study focused on specific machine learning algorithms: Random Forest and k-Nearest Neighbors, demonstrating comparable performance under different input data configurations in terms of compliance metrics such as R-squared, RMS error (RMSE) and correlation coefficient (r) between observed and calculated plant height values. A comparative analysis of the field measurements of plant height with the results obtained using machine learning methods was carried out. HotMap-diagrams are constructed showing the effectiveness of models in estimating the height of maize plants. Because of the study, it was possible to effectively demonstrate the applicability of images from unmanned aerial vehicles (UAV), orbital remote sensing and various machine learning algorithms for accurate assessment of plant height in the field. Moreover, the obtained results emphasize the feasibility of using an UAV with an RGB camera as a promising alternative for manually assessing the height of plants in the field. The relevance of the research is beyond doubt, but the authors have not disclosed it enough in their work. Recently, the use of aerial monitoring using both unmanned aerial vehicles and satellites, along with complex machine learning algorithms, has become increasingly used in modern agricultural systems. The article is well structured, with a clearly described methodology, the main part, and the results. The authors present the results of their experiments in a clear and understandable form, with tables and figures that effectively illustrate the main conclusions. The presented graphs and diagrams allow to visually evaluating the results of the experiment. The references, which includes 64 sources, effectively prepares the ground for the authors' own research and describes in detail the problematic situations related to this topic. High originality of the text of the article, which shows the high personal contribution of the authors. Thank you for your detailed and insightful review of our article, "Integrating Satellite and UAV Technologies for Maize Plant Height Estimation through Advanced Machine Learning." Your comprehensive summary of the key elements, from the purpose and methodology to the specific focus on machine learning algorithms and the comparative analysis, reflects a keen understanding of our research. We appreciate your acknowledgment of the relevance of our work in the context of modern agricultural systems and the increasing use of aerial monitoring with UAVs and
satellites. Your observation about the article's structure, clarity, and the effectiveness of visual aids like tables, figures, graphs, and diagrams is particularly gratifying. It's our aim to present our findings in a way that is not only scientifically rigorous but also accessible to a wide audience. Thanks again. General comments on the article. 1) It is necessary to better reveal the relevance of the research and show the scientific novelty of the work. We appreciate your feedback. We've implemented some changes and hope they have addressed your concerns. We are committed to further enhancing the quality of the paper and welcome any additional suggestions.2) To conduct a more reasoned justification of the selected machine learning methods, the reasons for choosing these methods. Likewise, between lines 188 and 191, our objective was to elucidate the reasoning behind the selection of algorithms employed in this study.3) Make a comparison with similar research; provide references to them. Thank you for the insightful suggestion. We share the belief that comparing our study with others utilizing machine learning algorithms and digital surface models will bolster its credibility. We have indeed conducted comparisons between our study’s approach and those of other researchers who utilized satellite and UAV platforms. 4) There are many abbreviations in the text that will be incomprehensible to readers who are not familiar with the topic of the research. It is necessary to provide a transcript at the first mention of abbreviations. Thank you once more for your comment. We’ve taken note of the numerous abbreviations throughout the text. Adjustments have been implemented, and we aspire to fulfill the request. Despite the comments, the study is a valuable contribution to the field of studying aerial monitoring using UAVs and satellites in agricultural systems, as the authors’ work is well done and the results are clearly presented. The article notes the transformative potential of combining satellite imagery, unmanned aerial vehicle technologies and machine learning for precision farming and estimating the height of maize plants. The presented technological approach provides valuable opportunities for both producers and researchers to improve monitoring of the dynamics of maize crop growth. Thank you for recognizing the value of our study in the field of aerial monitoring using UAVs and satellites in agriculture. We're thrilled that you see our work as a meaningful contribution, and your positive feedback on the clarity of our results is truly appreciated. It's heartening to know that our exploration of combining satellite imagery, UAV technologies, and machine learning for precision farming has been acknowledged for its transformative potential.

Reviewer 2 Report
Comments and Suggestions for Authors
1. This article uses mature algorithms, how can the innovative points be reflected?
2. The results evaluated in this article should be easily verifiable, why have they not been validated with experimental data?
3. Currently, not all agricultural plant protection drones in the market have adopted similar technologies. What is the scientific value of this study?
1. This article uses mature algorithms, how can the innovative points be reflected?
2. The results evaluated in this article should be easily verifiable, why have they not been validated with experimental data?
3. Currently, not all agricultural plant protection drones in the market have adopted similar technologies. What is the scientific value of this study?
Author Response
#REVIWER 2: Comments and Suggestions for Authors
1. This article uses mature algorithms; how can the innovative points be reflected? Thank you for the valuable suggestion. Although the algorithms employed in this study are well-established, their recent adaptation for estimating corn plant height in field conditions is noteworthy. These algorithms have showcased significant potential, reinforcing their utility as a tool adaptable for ongoing use in corn fields necessitating periodic assessments of plant height, crucial for monitoring the crop's growth and development. 2. The results evaluated in this article should be easily verifiable, why have they not been validated with experimental data? Thank you very much for your question. Model validation was not conducted for the study's results, as the K-fold method was utilized. This method involves training on randomly selected subsets of data, effectively mitigating issues related to overfitting and underfitting. Not only does this approach bolster improved learning within the model, but it also enhances its overall accuracy. 3. Currently, not all agricultural plant protection drones in the market have adopted similar technologies. What is the scientific value of this study? This research employs a suite of tools leveraging machine learning algorithms and UAV-captured images to estimate plant height within natural settings. This methodology presents an intriguing and cost-effective alternative when juxtaposed with LiDAR systems. The swift, precise, and cost-efficient estimation of plant height not only facilitates ongoing investigations but also lays the groundwork for future inquiries concerning canopy volume estimation. This, in turn, aids in determining more optimal spraying volumes for agricultural purposes.

Reviewer 3 Report
Comments and Suggestions for Authors
The work was performed on a current topic and is devoted to the automation of manual processes. Timely receipt of information about the development of plants growing over large areas allows not only to reduce labor costs, but also to make timely impacts. The results may also allow for early yield planning and planning of grower activities.
There are a number of comments to the presented material:
1. Astors did not clearly describe the relevance of the research, the significance of the application of the proposed technology and the advantages of manufacturers.
2. The required accuracy of the studied models is not justified
3. The conclusion should be specified based on the purpose and objectives of the study.
4. The advantages of the proposed method compared to the regulated ones are not presented.
Author Response
#REVIWER 3: The work was performed on a current topic and is devoted to the automation of manual processes. Timely receipt of information about the development of plants growing over large areas allows not only to reduce labor costs, but also to make timely impacts. The results may also allow for early yield planning and planning of grower activities. We ate glad you found the work relevant, especially given its focus on automating manual processes in a field as critical as plant development over large areas. The potential benefits you highlighted, such as the reduction of labor costs and the ability to make timely interventions, resonate with our objectives. We believe that the timely receipt of information can indeed revolutionize the way we approach plant development, enabling early yield planning and strategic grower activities. Your acknowledgment of these aspects encourages us in our pursuit of more efficient and effective methodologies in agriculture. There are a number of comments to the presented material: 1. Astors did not clearly describe the relevance of the research, the significance of the application of the proposed technology and the advantages of manufacturers.
We appreciate the suggestions, which will undoubtedly enrich the study. We concur on the significance of discussing the application's relevance and its advantages. In line 69, we emphasized the importance of plant height estimation through machine learning and remote sensing. We remain receptive to additional suggestions. 2. The required accuracy of the studied models is not justified Thank you very much for your question. The Random Forest and K-Nearest Neighbor emerged as the most effective machine learning algorithms, based on the statistical metrics R2 and RMSE. Furthermore, in line 325-335 and 448-454, we endeavored to elucidate the rationale behind their superior performance. We trust we have met your expectations and are keen to continue improving. 3. The conclusion should be specified based on the purpose and objectives of the study. Once more, we appreciate your attention. It is indeed crucial for the conclusion to align with this suggestion. Hence, we have revised the conclusion to ensure a clearer and more direct alignment with the study's objectives and purpose. 4. The advantages of the proposed method compared to the regulated ones are not presented. In the introduction section, our aim was to elucidate the benefits of employing vegetation indices integrated with machine learning to estimate such a crucial agronomic variable. We trust that these changes have enhanced readability and comprehension. Should you have any further inquiries, please feel free to reach out.

Reviewer 4 Report
Comments and Suggestions for Authors
Report Reviewer
Title: Integrating Satellite and UAV Technologies for Maize Plant Height Estimation through Advanced Machine Learning
Summary brief
This study emphasises the significant transformative potential of combining satellite imagery, unmanned aerial vehicle (UAV) technology and machine learning within precision agriculture and maize plant height estimation. The main aim is to evaluate the accuracy of high-resolution satellite imagery, integrated with an RGB camera mounted on a UAV, for estimating the height of maize plants by exploiting advanced machine learning algorithms. In the course of the study, three vegetation indices were calculated at different growth stages of maize plants. The results of the study show that the height of maize plants, obtained from DEMs, shows a strong correlation with manual measurements in the field and the Random Forest (RF) model is the best, for estimating height values.
General concept comments
The manuscript is well done and arouses high scientific interest. The English language was used correctly and made the work easy to read. The introduction manages to convey what is known in the literature and the gaps in knowledge to introduce the aim of the work. The in-depth description of the materials and methods used so far in the literature was much appreciated. The methodology and statistics were correctly applied, although a few more statistical analyses could have strengthened the results. The discussion, despite its length, was also well illustrated. The data were presented in a linear and comprehensive manner, although sections 3.1 and 3.3 could be unified as they show identical results. Therefore, I do not think there is much revision to be done. Below are some suggestions
- Row 102: Enter the BBCHs coinciding with the survey days.
- Row 128: Were reflectance calibration panels placed? If yes, please indicate the methodology used.
- Results: The results show several repeated values and show the same thing. What is the difference between paragraph 3.1 and 3.3? It would probably be better to merge the two paragraphs into one.
Comments on the Quality of English LanguageReport Reviewer
Title: Integrating Satellite and UAV Technologies for Maize Plant Height Estimation through Advanced Machine Learning
Summary brief
This study emphasises the significant transformative potential of combining satellite imagery, unmanned aerial vehicle (UAV) technology and machine learning within precision agriculture and maize plant height estimation. The main aim is to evaluate the accuracy of high-resolution satellite imagery, integrated with an RGB camera mounted on a UAV, for estimating the height of maize plants by exploiting advanced machine learning algorithms. In the course of the study, three vegetation indices were calculated at different growth stages of maize plants. The results of the study show that the height of maize plants, obtained from DEMs, shows a strong correlation with manual measurements in the field and the Random Forest (RF) model is the best, for estimating height values.
General concept comments
The manuscript is well done and arouses high scientific interest. The English language was used correctly and made the work easy to read. The introduction manages to convey what is known in the literature and the gaps in knowledge to introduce the aim of the work. The in-depth description of the materials and methods used so far in the literature was much appreciated. The methodology and statistics were correctly applied, although a few more statistical analyses could have strengthened the results. The discussion, despite its length, was also well illustrated. The data were presented in a linear and comprehensive manner, although sections 3.1 and 3.3 could be unified as they show identical results. Therefore, I do not think there is much revision to be done. Below are some suggestions
- Row 102: Enter the BBCHs coinciding with the survey days.
- Row 128: Were reflectance calibration panels placed? If yes, please indicate the methodology used.
- Results: The results show several repeated values and show the same thing. What is the difference between paragraph 3.1 and 3.3? It would probably be better to merge the two paragraphs into one.
Author Response
#REVIWER 4: Summary brief This study emphasises the significant transformative potential of combining satellite imagery, unmanned aerial vehicle (UAV) technology and machine learning within precision agriculture and maize plant height estimation. The main aim is to evaluate the accuracy of high-resolution satellite imagery, integrated with an RGB camera mounted on a UAV, for estimating the height of maize plants by exploiting advanced machine learning algorithms. In the course of the study, three vegetation indices were calculated at different growth stages of maize plants. The results of the study show that the height of maize plants, obtained from DEMs, shows a strong correlation with manual measurements in the field and the Random Forest (RF) model is the best, for estimating height values. General concept comments The manuscript is well done and arouses high scientific interest. The English language was used correctly and made the work easy to read. The introduction manages to convey what is known in the literature and the gaps in knowledge to introduce the aim
of the work. The in-depth description of the materials and methods used so far in the literature was much appreciated. The methodology and statistics were correctly applied, although a few more statistical analyses could have strengthened the results. The discussion, despite its length, was also well illustrated. The data were presented in a linear and comprehensive manner, although sections 3.1 and 3.3 could be unified as they show identical results. Therefore, I do not think there is much revision to be done. Below are some suggestions Thank you for your thoughtful and constructive feedback on our manuscript. We appreciate your positive comments on the overall quality of the work, as well as your recognition of the clarity in language and the effective use of the introduction to set the stage for our study. We acknowledge your suggestion regarding additional statistical analyses to strengthen the results. We will carefully consider this and explore potential areas where supplementary analyses could enhance the robustness of our findings, however we belive the statistical approach used is enough to explain our approach. Your observation about unifying sections 3.1 and 3.3 is duly noted, and we worked on streamlining those sections to avoid redundancy. We are grateful for your time and insights, and we are committed to making the necessary revisions to improve the manuscript. Your feedback is invaluable, and we look forward to hear your acceptance after an enhanced version of our work based on your recommendations. - Row 102: Enter the BBCHs coinciding with the survey days. Thank you for your valuable comment, which will significantly enhance the study. We were unsure about the meaning of BBCHs. Nonetheless, striving to enhance the content in line 102, we included the dates when the satellite and UAV images were collected. We hope this modification aligns with your suggestion. - Row 128: Were reflectance calibration panels placed? If yes, please indicate the methodology used. Thank you. No reflectance calibration panel was utilized. The vegetation indices utilized in the model were derived from orbital images acquired through the PlanetScope platform. This platform offers a product incorporating geometric and radiometric correction, thus obviating the necessity of a calibration panel. Known as PlanetScope Analytic Ortho Scene Surface Reflectance (SR-Level 3B), this product ensures refined data quality. Concurrently, images captured by the UAV were employed for constructing digital elevation models. - Results: The results show several repeated values and show the same thing. What is the difference between paragraph 3.1 and 3.3? It would probably be better to merge the two paragraphs into one. We appreciate your suggestion and attention. Indeed, there were duplicated pieces of information regarding the algorithms, and we acknowledge the necessity of merging sections 3.1 and 3.3. As a result, we have made the alteration and hope it meets your request. Thank you once again for your thorough review.

Round 2
Reviewer 2 Report
Comments and Suggestions for Authors
The discussion section can be more detailed and in-depth.
Comments on the Quality of English LanguageNo.